# Achieving Endo/Lysosomal Escape Using Smart Nanosystems for Efficient Cellular Delivery

**DOI:** 10.3390/molecules29133131

**Published:** 2024-07-01

**Authors:** Nimeet Desai, Dhwani Rana, Sagar Salave, Derajram Benival, Dignesh Khunt, Bhupendra G. Prajapati

**Affiliations:** 1Indian Institute of Technology Hyderabad, Kandi 502285, Telangana, India; nimeet.desai@gmail.com; 2National Institute of Pharmaceutical Education and Research (NIPER), Ahmedabad 382355, Gujarat, India; dhwanirana73@gmail.com (D.R.); sagarsalave1994@gmail.com (S.S.); derajram@niperahm.res.in (D.B.); 3School of Pharmacy, Gujarat Technological University, Gandhinagar 382027, Gujarat, India; 4Shree S. K. Patel College of Pharmaceutical Education and Research, Ganpat University, Kherva 384012, Gujarat, India; 5Faculty of Pharmacy, Silpakorn University, Nakhon Pathom 73000, Thailand

**Keywords:** endosomal escape, nanoparticles, cellular delivery, smart nanomaterials, biomolecule delivery, nanomedicine

## Abstract

The delivery of therapeutic agents faces significant hurdles posed by the endo-lysosomal pathway, a bottleneck that hampers clinical effectiveness. This comprehensive review addresses the urgent need to enhance cellular delivery mechanisms to overcome these obstacles. It focuses on the potential of smart nanomaterials, delving into their unique characteristics and mechanisms in detail. Special attention is given to their ability to strategically evade endosomal entrapment, thereby enhancing therapeutic efficacy. The manuscript thoroughly examines assays crucial for understanding endosomal escape and cellular uptake dynamics. By analyzing various assessment methods, we offer nuanced insights into these investigative approaches’ multifaceted aspects. We meticulously analyze the use of smart nanocarriers, exploring diverse mechanisms such as pore formation, proton sponge effects, membrane destabilization, photochemical disruption, and the strategic use of endosomal escape agents. Each mechanism’s effectiveness and potential application in mitigating endosomal entrapment are scrutinized. This paper provides a critical overview of the current landscape, emphasizing the need for advanced delivery systems to navigate the complexities of cellular uptake. Importantly, it underscores the transformative role of smart nanomaterials in revolutionizing cellular delivery strategies, leading to a paradigm shift towards improved therapeutic outcomes.

## 1. Introduction

The advancement of therapeutic approaches has progressed beyond traditional small-molecule drugs towards a diverse array of innovative approaches such as proteins, peptides, monoclonal antibodies, nucleic acids, and live cells [1,2,3,4,5,6,7,8,9]. Accordingly, corresponding advancements in drug delivery technologies have emerged to address the distinct requirements of these novel therapeutics [10]. This extends to gene delivery, where innovative systems such as viral vectors, nanoparticles, and targeted strategies are vital for effective delivery of nucleic acid therapeutics to target cells and tissues, enabling precision interventions aligned with individual genetic characteristics. While viral vectors have traditionally been favored for gene delivery due to their high transfection efficiency, their utilization also carries the inherent risk of eliciting immune responses and inducing aberrant insertional mutagenesis. Consequently, there has been a notable expansion in research exploring alternative non-viral methods for gene delivery [11]. Nanoparticles hold great promise for improving therapeutic delivery by protecting contents, targeting specific cells or tissues, and regulating content release within desired cellular regions [12,13,14,15,16,17,18,19,20,21,22]. They are particularly important for delivering biological therapeutics like DNA, RNA, or proteins, which face stability issues when administered alone. Precision cell-based therapies and fundamental biomedical research often require delivering therapeutics to specific intracellular compartments. Despite advancements in encapsulation and release, there are still hurdles to achieving efficient intracellular delivery. The majority of nanoparticle delivery systems are internalized into cells through the endocytosis pathway [23].

Given that most drug delivery systems traverse the endosome–lysosome pathway upon cellular uptake, the efficient delivery of therapeutic drugs relies heavily on the ability of drug delivery systems to escape the endosome-lysosome pathway post-cellular-uptake [24,25]. Most intracellular transport pathways involve several stages, including internalization into an endocytic vesicle, fusion with the early endosome (EE), maturation into a late endosome (LE), and eventual accumulation in the lysosome (Figure 1). Throughout this process, the pH gradually decreases from approximately 7.4 to around 5.0 within the lysosome, which contains various degradative enzymes. Failure to escape rapidly from lysosomes can lead to entrapment and potential degradation of therapeutic drugs, rendering the delivery unsuccessful. Furthermore, many nanocarriers face lysosomal degradation after internalization by endothelial cells via endocytosis, thus failing to penetrate the blood–brain barrier (BBB) through transcytosis [24]. Overcoming the endosomal/lysosomal barrier is essential for successful nanodrug delivery in the treatment of various diseases. This article outlines the significance of endosomal entrapment, emphasizing its role as a barrier to drug/biomolecule delivery and the need for “smart” nanosystems to overcome it. Assays to study endosomal escape/cellular uptake including leakage assays, complementation assays, cytosolic-activation assays, pharmacologic/genetic screens, and co-localization studies have also been presented.

Understanding the mechanisms that regulate endosomal escape is essential for the rational design of efficient and nontoxic delivery systems. Various strategies have been explored to achieve rapid drug release into the cytoplasm, aiming to address these challenges. This review highlights strategies based on biological mechanisms such as pore formation, the proton sponge effect, membrane destabilization, photochemical disruption, and the use of endosomal escape agents. By incorporating these strategies, novel and comprehensive approaches can be developed to overcome endosomal/lysosomal barriers and enhance the efficiency of nanotherapeutic delivery. Smart nanosystems present a promising strategy to the challenge of endosomal entrapment in drug delivery, employing engineered nanocarriers designed to respond to the cellular environment and facilitate escape from endosomes. Further, this review examines the challenges encountered by smart nanosystems as they advance towards clinical approval, with the goal of providing guidance for the informed design of next-generation nanomedicines for clinical utilization.

## 2. Significance of Endosomal Entrapment

The plasma membrane of a cell functions dynamically as a barrier, managing the flow of biomolecules between its interior and exterior. Integral membrane proteins like channels and pumps facilitate the passage of small molecules such as sugars and ions. However, larger macromolecules necessitate internalization via primary endocytic vesicles (PEVs), which form through invaginations of the plasma membrane. These vesicles deliver their contents to EEs in the peripheral cytoplasm. The process of uptake and transporting extracellular material within membrane-bound vesicles is collectively referred to as endocytosis. Endosomes and lysosomes are essential membrane-bound organelles vital for the proper functioning of eukaryotic cells. The specifics of endocytic pathways vary depending on the size of the particles being internalized and the scale of the invaginations they form. For instance, micropinocytosis encompasses invaginations smaller than 200 nm and involves both clathrin-mediated (CME) and non-clathrin-mediated endocytosis (NCE) pathways. In CME, clathrin-coated vesicles form to internalize receptor–ligand complexes from the plasma membrane. Molecules taken up through CME include hormones, transferrin, and low-density lipoprotein, along with their respective receptors. NCE encompasses various pathways, such as clathrin-independent fluid-phase endocytosis, used for sampling the extracellular environment, and caveolar endocytosis, which is implicated in the uptake of certain viruses and sphingolipids. Particles larger than 500 nm, such as bacterial pathogens and apoptotic cell debris, are typically engulfed through phagocytosis, a process occurring in specialized cells of the innate immune system. Particles sized between 200 and 500 nm are internalized through macropinocytosis, a mechanism involving large-scale rearrangements of the plasma membrane [27,28,29].

Upon internalization, the molecules are conveyed to EEs, serving as the principal sorting center within the endocytic pathway. Biological cargoes internalized through various endocytic pathways are initially directed to the EE, where the interior pH maintains slightly acidic level, around 6.3. As this EE matures into the LE, there is a subsequent decline in pH to around 5.5. Ultimately, fusion occurs between the LE and the lysosome, which boasts a pH of approximately 4.7. This leads to the degradation of the cargoes, facilitated by hydrolytic enzymes. Some receptors and specific proteins have the capability of being recycled back to the plasma membrane, while other molecules—including downregulated receptors—are transported to late endosomes and lysosomes for degradation [30].

Endosomal entrapment serves as a natural mechanism within cells for regulating intracellular trafficking and controlling the fate of internalized materials. It plays a pivotal role in sorting and directing molecules to their appropriate destinations within the cell, such as recycling back to the cell surface, degradation within lysosomes, or transport to other intracellular compartments. This inherent mechanism ensures proper cellular function and homeostasis. However, from a therapeutic perspective, endosomal entrapment poses a significant challenge. Many therapeutic agents, including drugs, proteins, and nucleic acids, are internalized by cells through endocytosis but become sequestered within endosomes. This entrapment restricts their bioavailability and efficacy, preventing them from reaching their intended intracellular targets. Thus, while endosomal entrapment is an essential physiological process for cellular function, overcoming it, or achieving endosomal escape, becomes crucial for enhancing the intracellular delivery of therapeutic agents and maximizing their therapeutic potential. This necessitates the development of innovative drug delivery systems that can navigate through endosomal barriers and efficiently release cargo into the cytoplasm, where it can exert its desired effects.

Smart nanomaterial-based delivery systems have gained considerable attention to efficiently deliver cargo in a targeted manner and surpass the biological barriers upon administration [4,7,8,9,19,31]. These nanosystems comprise two main components: a nanomaterial support or carrier and a functionalizing biomolecule [32]. Smart nanocarriers are engineered with unique properties that enable them to respond to the cellular environment and facilitate escape from endosomes [33].

Smart nanoparticles can modify their form, structure, surface charge, solubility, self-association, or dissociation behaviors in response to internal and external stimuli. This dynamic adaptability enables them to improve endosomal escape, promote cellular uptake, and induce payload release [34]. Various nanosystems responsive to stimuli have been developed to exploit pathological differences for targeting and delivering cargoes within the cytoplasm [35]. One approach involves pH-responsive nanocarriers, taking advantage of the acidic pH environment within organelles (such as lysosomes and endosomes) of cancer cells and in the tumor microenvironment. Typically, the pH in the cytoplasm, blood, and normal tissues hovers around pH 7.0 to 7.4, whereas it drops to approximately pH 6 to 4 in endosomal/lysosomal organelles and pH 6.5 to 6.8 in the tumor microenvironment. Consequently, pH-responsive systems in the tumor microenvironment can be used for controlled drug release or prodrug activation while maintaining the stealth effect of nanocarriers in normal regions (e.g., in the bloodstream) to prevent cargo leakage. This approach reduces the risk of exposing normal organs (e.g., the heart) to toxic cargoes (e.g., doxorubicin) and specifically delivers them to tumors, thereby enhancing therapeutic efficacy. Additionally, several pH-sensitive polymers have been synthesized for constructing nanocarriers with pH responsiveness [36]. Another approach focuses on leveraging photothermal activation to induce endosomal escape, utilizing light-responsive materials, photoactivatable molecules, or nanocarriers equipped with photothermal or photodynamic agents. For instance, a photothermally triggered system has been developed for targeted delivery of siRNA into the cytoplasm using ultrasmall melanin nanoparticles triggered by near-infrared (NIR) irradiation. The nanoparticles consist of a melanin-poly-L-lysine (M-PLL) polymer, where melanin serves as a photothermal sensitizer, and PLL condenses siRNA through electrostatic interactions, enabling “on-demand” endosomal escape and controlled release of siRNA, thus enhancing therapeutic efficacy [37].

Functionalized nanoparticles are integral to smart nanosystems, especially in addressing the challenge of endosomal escape for efficient intracellular delivery. By incorporating peptides such as cell-penetrating peptides (CPPs) or lysosomal sorting peptides, nanoparticles achieve precise cellular targeting [38,39,40]. Additionally, nanocarriers equipped with specific antibodies offer an approach to reaching desired cellular destinations [41]. The inclusion of fusogenic lipids and peptides further enhances nanoparticle functionality by facilitating membrane translocation and enhanced endosomal escape [42]. An illustrative example of this is the demonstration that pH-sensitive fusogenic GALA peptides promote endosomal escape from a bio-nanocapsule through an endocytic uptake pathway [43,44]. Furthermore, functionalizing the surface of nanoparticles using various polymers presents an advantageous approach capable not only of manipulating the nanoparticles’ characteristics—including size, shape, charge, smoothness, hydrophilicity/hydrophobicity, homogeneity, and stability—but also of serving as a driving force for enhancing cellular internalization, facilitating endosomal escape, and optimizing the drug release profile, ultimately aiming to achieve subtle therapeutic effects [45]. This strategic integration of various functional components highlights the potential of functionalized nanoparticles in optimizing intracellular delivery mechanisms within smart nanosystems.

## 3. Assays to Study Endosomal Escape/Cellular Uptake

Evaluating endosomal escape and uptake mechanisms is imperative for comprehending the effectiveness of drug delivery systems and articulating strategies to enhance intracellular delivery. As explained previously, endosomal escape initiates with membrane destabilization, followed by pore formation, endosomal rupture, or membrane fusion [46]. Typically, investigations into endosomal escape mechanisms focus on assessing the integrity of the endosomal membrane. The development of nanocarriers demonstrating escape activity encounters formidable challenges due to the lack of efficient techniques for detecting/quantifying endosomal escape events. This impedes the delineation of material attributes fostering escape capability and the identification of potential modifications to enhance it [47]. It is worth noting that there is no standardized technique for elucidating intracellular trafficking or endosomal escape mechanisms. Rather, researchers rely on various experimental methodologies (or a combination thereof) to shed light on distinct facets of the process. At present, methods for evaluating endosomal escape can be categorized into the following groups: leakage assays, complementation assays, cytosolic-activation assays, and pharmacological/genetic screens.

### 3.1. Leakage Assays

Leakage assays represent straightforward techniques utilized to identify disruptions within endosomal membranes, relying on the detection of fluorescent dyes or other discernible molecules outside of the enclosed compartment. A seminal investigation in this domain was conducted by Manganiello et al. [48], who employed an in vitro hemolysis assay to elucidate the membrane-disruptive potential of a diblock copolymer micelle. In this study, free polymer entities were incubated with erythrocytes in buffers with varying pH levels to simulate the acidic milieu of the endosome. The quantification of hemoglobin released, assessed through absorbance measurements, served as a metric for evaluating the extent of membrane disruption. Similarly, another leakage-based strategy entailed the use of formulated vesicles, mimicking endosomal composition, as an alternative to erythrocytes to closely replicate escape dynamics [49]. Nonetheless, these methodologies may fall short in adequately replicating the intricate environment of the acidifying endosome within the cellular context, potentially yielding results that diverge from actual cellular processes [50].

To circumvent this limitation, researchers have devised more physiologically relevant approaches to leakage-based assays within cell culture settings. For instance, Su et al. [51] developed a study wherein the calcein was co-incubated with lipid-enveloped poly(β-amino ester) nanoparticles. Calcein, a membrane-impermeable fluorescent dye, is passively macropinocytosed by cells and transported into endosomes and lysosomes. Within these acidic compartments, calcein undergoes self-quenching at high concentrations and low pH levels, resulting in punctate fluorescence with diminished intensity when entrapped within endocytic vesicles. Conversely, upon membrane compromise, calcein diffuses into the cytosol, leading to augmented fluorescence. Flow cytometry enables the differentiation of distinct cell populations based on fluorescence intensity [52]. While leakage assays employing small molecules such as calcein offer a means to surmount the intrinsic low signal-to-noise ratio associated with detecting minimal release induced by nanoparticles, akin to most leakage assays, this method is constrained by the potential for dye leakage without the release of nanoparticles or cargo [53]. Additionally, this mode of escape assessment remains qualitative, merely ascertaining the occurrence of escape rather than quantifying its efficiency [54].

### 3.2. Complementation Assays

This method entails utilizing cytosol-expressed proteins to elicit a response upon interaction with the escaped cargo [55]. An illustrative case involves the glucocorticoid receptor (GR), which complexes with heat shock protein 90 (hsp90) within the cytosol. Upon exposure to the agonist dexamethasone (Dex), hsp90 disengages, facilitating receptor translocation to the nucleus [56]. This phenomenon has been leveraged to interrogate the cell-penetrating attributes of peptides and miniature cationic proteins. In a study by Appelbaum et al. [57], cells were genetically modified to express a GR-green fluorescent protein (GFP) fusion construct, yielding a diffuse GFP signal throughout the cytosol and nucleus. Prospective escape candidates were conjugated with Dex and co-incubated with the transfected cells. Successful cytosolic ingress prompted nuclear translocation and concomitant augmentation of GFP fluorescence within this compartment. Utilizing fluorescence microscopy, the nuclear-to-cytosolic GFP signal ratio served as a relative metric of escape proficiency. Nonetheless, the GFP in this assay invariably emits fluorescence and merely undergoes relocation, confining its applicability to microscopy (low-throughput).

To surmount this limitation, an assay where a signal manifests solely post-escape proves advantageous, exemplified by the split-complementation assay. Herein, reporter proteins are bifurcated into two non-functional fragments, generating a detectable output exclusively upon their reconstitution. Primarily employed to probe protein–protein interactions, subcellular protein localization, and protein assembly, this assay paradigm offers versatility [58]. Milech et al. [59] devised a tailored split GFP complementation assay to quantify the endosomal escape of a library of cell-penetrating peptides (CPPs) fused to cargo proteins. Initially, cells were transfected with one half of the GFP protein, ensuring stable expression of the inert fragment in the cytosol. Subsequently, the other GFP moiety was affixed to the cargo-CPP fusion. Upon CPP-mediated endosomal escape, the two GFP fragments coalesced, generating a fluorescent signal. This assay boasts minimal background noise and furnishes a direct readout independent of enzymatic processes. Moreover, its protocol lends itself to relatively high-throughput implementation facilitated by automated microscopy, flow cytometry, or plate readers. Analogous protein complementation assays have been extended to assess the escape of alternative carriers, including cationic lipids and polyplexes [60].

A variant of the GFP complementation assay directly quantifies endosomal disruption. Kilchrist et al. [61] devised two split-luciferase assays predicated on galectin 8 (Gal8) protein interactions recruited to compromised endosomes and lysosomes. In one assay, Gal8 is fused to an N-terminal luciferase fragment, while CALCOCO2 binds to the C-terminal fragment. Following endosomal disruption, Gal8 relocates intracellularly and recruits CALCOCO2, reuniting the luciferase fragments to form a functional enzyme with luminescent output. The second assay exploits Gal8 dimerization to juxtapose the luciferase fragments. This methodology was validated for liposomal- and polymer-based delivery systems, albeit limited sensitivity curtailed its in vivo applicability. While this approach offers rapid, quantitative assessment sans the need for carrier or cargo labeling, it solely discerns endosomal disruption and does not ascertain cargo escape.

### 3.3. Cytosolic Activation Assays

These assays exploit the principle of utilizing pro-drug substrates, pH-sensitive probes, or inactive enzymes encapsulated within nanoparticles. Upon successful endosomal escape and subsequent release into the cytosol, these entities undergo activation, thereby eliciting discernible cellular responses [62]. For instance, certain fluorescent probes exploit the disparity in pH levels between the acidic endosomal environment and the neutral cytosol as an indicator of successful escape. These probes may be co-incubated with the carrier vehicles or directly conjugated to polymeric or micellar carriers [63,64].

Jiang et al. [65] pioneered the use of a deglycosylation-dependent Renilla luciferase (ddRLuc) probe engineered with crucial amino acid substitutions rendering it enzymatically inert. This probe can be encapsulated within the investigated carrier vehicle, and upon successful endosomal escape, the cytosolic enzyme N-glycanase-1 (NGLY1) activates the luciferase. In instances in which the vehicle was loaded with mRNA, this release assay demonstrated a direct correlation with in vitro mRNA transfection efficiency. Another iteration of the cytosolic activation concept, closely mimicking gene delivery mechanisms, is the splicing reporter system developed by Guterstam et al. Here, a HeLa cell line was stably transfected with a non-functional luciferase harboring an aberrant splice-site. Subsequently, a complementary oligonucleotide capable of masking the splice-site was delivered, facilitating the production of functional luciferase. The activity of the produced luciferase was quantifiable using a luminometer, providing valuable insights into the efficiency of endosomal escape and subsequent cytosolic activation [66].

Figure 2 provides a schematic overview of the above-discussed assays that can be implemented to assess endosomal escape.

### 3.4. Pharmacologic/Genetic Screens

Pharmacological/genetic screens involve deliberately applying pharmacological agents or genetic manipulations to modulate specific pathways or molecular targets associated with intracellular trafficking and endosomal escape. Through careful observation of the resulting effects on cellular uptake and intracellular distribution of nanoparticles, these screens offer invaluable insights into the molecular mechanisms governing these processes [67,68]. Pharmacological inhibitors have historically served as fundamental tools in elucidating endocytosis and intracellular trafficking mechanisms. By selectively disrupting particular pathways, researchers can discern which processes and molecules are indispensable for efficient intracellular delivery [69,70].

Chlorpromazine, hypertonic sucrose, and potassium depletors are among the common inhibitors of clathrin-mediated endocytosis (CME), whereas cholesterol depletors like statins or methyl-β-cyclodextrin inhibit caveolae-mediated endocytosis [71]. Researchers have also employed inhibitors to investigate specific hypotheses regarding endosomal escape mechanisms. For instance, Kichler et al. [72] tested the proton sponge hypothesis for PEI-mediated delivery by using proton pump inhibitors, bafilomycin A1, and concanamycin A, to assess any reduction in endosomal escape. The inhibition strategy lends itself well to high-throughput studies, exemplified by Sahay et al.’s screening of a library of small molecule inhibitors in cell culture, where microscopy was employed to identify the effectors necessary for lipid nanoparticle cellular entry [73]. However, it is noteworthy that most inhibitors may exert effects on multiple intracellular processes, thus limiting the specificity of this approach [74]. Additionally, studies have indicated that the effects of chemical inhibitors can vary depending on the cell line used, adding a layer of complexity to their interpretation [75]. Table 1 provides a concise overview of various pharmacological inhibitors available for facilitating the endosomal escape of nanosystems.

On the genetic front, advanced techniques like RNA interference (RNAi) or CRISPR/Cas9-mediated gene knockout offer powerful tools for selectively manipulating genes implicated in endosomal trafficking and membrane dynamics [78,79]. Panarella et al. [80] devised two siRNA libraries targeting pertinent cytoskeletal and endosomal genes. Employing an automated high-throughput microscopy protocol, they assessed the impact on nanoparticle delivery. Similarly, Ross-Thriepland et al. [81] conducted a screening experiment, employing the CRISPR/Cas9 gene editing platform to investigate lipid nanoparticle-mediated mRNA delivery. Utilizing a pooled design, they scrutinized 7795 genes, identifying 44 hits that either enhanced or diminished transfection efficiency. Despite the promise of CRISPR/Cas9-based approaches, the potential for off-target effects necessitates additional validation steps [82]. Furthermore, certain endosomal proteins participate in multiple trafficking and cellular processes, complicating the interpretation of results. Like pharmacological inhibitor screens, genetic screens cannot unequivocally attribute observed effects to a single target or pathway [83]. Additionally, gene trapping has emerged as a valuable technique for elucidating mechanisms of vesicle trafficking and viral infection. However, its application in the context of non-viral delivery remains largely unexplored [84,85]. Expanding the repertoire of genetic screening methods holds promise for uncovering novel insights into the intricate processes governing intracellular trafficking and endosomal escape, thereby advancing the development of more effective drug delivery strategies.

### 3.5. Other Techniques

#### 3.5.1. Co-Localization Studies

Co-localization studies serve as a fundamental methodology in unraveling the intricate mechanisms governing intracellular trafficking and endosomal escape within drug delivery systems. These studies entail the visualization and quantification of the spatial overlap between nanoparticles and intracellular organelles, particularly endosomes and lysosomes, leveraging appropriate imaging techniques. By monitoring the temporal progression of co-localization events, researchers can discern the rate at which nanoparticles are internalized by cells, trafficked through the endocytic pathway, and potentially liberated into the cytoplasm [86].

Immunohistochemistry and fluorescence microscopy are commonly employed in co-localization studies to enable detailed observations of nanocarrier’s intracellular trafficking [87]. Dyes, akin to those utilized in leakage assays, discern specific intracellular compartments [88], while immunofluorescence staining labels proteins like EEA1 and LAMP1, indicative of vesicle stages within the endolysosomal pathway [89]. Employing these methods alongside secondary dye-conjugated delivery vehicles, labeled nucleic acid cargo, and/or reporter nucleic acids, researchers systematically trace nanocarriers throughout intracellular pathways. In an illustrative study, nanoparticles loaded with coumarin-6 underwent screening for co-localization with approximately 30 Rab GTPase proteins alongside factors pertinent to clathrin-independent and clathrin-dependent uptake pathways [90]. Similar methodologies have previously probed the trafficking pathways of chitosan-based nanosystems [91]. However, while co-localization studies do not directly quantify endosomal escape or interactions with trafficking markers, the adoption of 3D laser scanning microscopy and live cell imaging augments spatial and temporal observations [33,92].

Fluorescence resonance energy transfer (FRET) offers an alternative avenue for visualizing molecular interactions between delivery vehicles and cellular constituents. Harnessing two fluorophores—a donor in an excited state transferring energy to an acceptor via long-range dipole coupling, FRET facilitates precise localization of delivery vehicles [93]. For instance, liposomes formulated with FRET labels Rho-PE and NBD-DOPE exhibited changes in fluorescence intensity ratio indicative of endosomal membrane fusion, enabling real-time observation of endosomal escape [94]. Use of computational image analysis aids in quantifying co-localization by correlating fluorescent signals from different channels, offering insights into nanoparticle distribution and cargo release kinetics [95].

Transmission electron microscopy (TEM) serves as a complementary technique to optical microscopy owing to its superior resolution. TEM discerns vehicle interactions with cellular structures, offering insights unattainable via fluorescence microscopy [96]. For instance, TEM imaging of lipid-nanoparticle-mediated siRNA delivery revealed a minute fraction of siRNA escaping endosomes, highlighting TEM’s utility in discerning subtle intracellular events [97]. However, TEM necessitates extensive sample preparation and electron-dense labels. Atomic force microscopy (AFM) complements TEM by mapping sample topography, offering insights into vehicle morphology and polyplex structure variations under cytosolic and endosomal conditions [98].

#### 3.5.2. Biologically Relevant Artificial Membranes

Artificial membranes, designed to mimic the complexities of the endosomal membrane, serve as pivotal tools in ex cellulo assays for assessing endosomal escape dynamics within a controlled environment [99]. The principal constituents of the endosomal membrane, including phosphatidylcholine (PC), phosphatidylethanolamine (PE), and phosphatidylserine (PS), typically constitute 55%, 25%, and 10% of the total lipid content, respectively [100,101]. Although simplistic models often comprise solely PC and occasionally cholesterol, recent endeavors aim to mirror the endosomal membrane composition more authentically [102]. The significance of lipid composition in membrane interactions has been underscored by studies such as Berezhna et al., wherein lipid fusion was scrutinized between lipoplexes and giant unilamellar vesicles of varying PC, PE, PS, and sphingomyelin (SM) compositions. Intriguingly, the authors delineate that fusion and subsequent nucleic acid release mediated by cationic lipoplexes predominantly hinge on the presence of negatively charged PS and PE, while PC and SM exhibit negligible involvement [103].

Similarly, Yang et al. [104] demonstrated the indispensability of the anionic lipid bis(monoacylglycero) phosphate, abundantly present in the intraluminal vesicles of late endosomes, for TAT-mediated fusion processes. These findings underscore the critical role of lipid composition in ex cellulo assays, emphasizing its pivotal impact on endosomal escape phenomena. Moreover, the endosomal environment’s distinctive acidic pH necessitates appropriate pH adjustments in artificial endosome resuspension buffers to replicate physiological conditions accurately. An alternative strategy proposed by Madani et al. [105] entails the integration of bacteriorhodopsin within artificial large unilamellar vesicles. Upon illumination, BR functions as a proton-pumping V-type ATPase, effectively acidifying the vesicles’ interior and mimicking the late endosomal milieu under controlled conditions. Nevertheless, it is imperative to acknowledge the inherent generalizations of liposomal models compared to actual endosomal membranes, primarily due to the absence of proteins and lipid asymmetry [106]. Despite these limitations, artificial membranes represent invaluable tools for elucidating fundamental aspects of endosomal dynamics and guiding the rational design of efficacious drug delivery systems.

## 4. Leveraging Smart Nanocarriers: Mechanism & Case Studies

### 4.1. Via Pore Formation

Numerous nanoparticles encounter challenges in achieving efficient cytosolic delivery due to entrapment within lysosomes. A straightforward strategy to facilitate escape involves co-treatment with or co-delivery of lysosomotropic agents, such as chloroquine (CQ), aimed at permeabilizing the lysosomal membrane [107]. These agents possess the ability to freely diffuse into lysosomes, whereupon protonation by the acidic milieu induces membrane disruption via pore formation [108]. Du Rietz et al. [109] devised a strategy to augment the transfection efficiency of cholesterol-conjugated siRNA by a remarkable 47-fold through the targeted delivery of CQ. Employing galectin-9 as a sensitive indicator for membrane disruption, they elucidated the precise mechanism by which chloroquine mediates endosomal escape. Their findings unequivocally demonstrate that CQ acts upon late endosomes or lysosomes, thereby facilitating efficient release of the siRNA payload into the cytoplasm. The process of pore formation by lysosomotropic agents relies on the intricate interplay between membrane tension, which promotes pore enlargement, and line tension, which counteracts pore expansion. Notably, certain components, such as peptides, exhibit a pronounced affinity for the pore rim. The binding of peptides to the pore rim reduces line tension, thereby stabilizing the pore radius and mitigating fluctuations in internal membrane tension [110].

Evidence suggests that the binding of molecules like cationic amphiphilic peptides (AMPs) to the lipid bilayer induces internal stress or membrane tension of sufficient magnitude to induce pore formation [111]. Several models, including the barrel-stave pore and toroidal pore models, have been proposed to elucidate the mechanisms underlying peptide-induced pore formation [112]. In the barrel-stave pore model, peptides reorient themselves to form staves, collectively assembling into a barrel-shaped cluster perpendicular to the lipid bilayer plane, thereby creating the pore [113]. Conversely, the toroidal pore model implicates peptide aggregates that insert into the membrane in a perpendicular orientation, inducing inward membrane curvature and ultimately resulting in pore formation, with peptides lining the pore interior [114,115].

Shao et al. [116] investigated the encapsulation of CQ and doxorubicin (DOX) using methoxy poly(ethylene glycol)-poly(L-lactic acid) (MPEG-PLA) nanoparticles to enhance anticancer effects in ovarian cancer. CQ significantly increased the sensitivity of ovarian cancer cells to chemotherapeutic agents by inhibiting lysosome-mediated drug sequestration, thus enhancing cytotoxicity (Figure 3). Through a series of in vitro and in vivo experiments, the authors demonstrated that pretreatment with CQ effectively reversed the sequestration of DOX by lysosomes, leading to increased drug accumulation within tumor cells and subsequent enhancement of cytotoxicity. Furthermore, the co-encapsulation of CQ and DOX within MPEG-PLA nanoparticles resulted in improved drug delivery efficiency, as evidenced by enhanced drug accumulation at tumor sites and prolonged circulation time in vivo. The MPEG-PLA nanoparticles exhibited favorable characteristics, such as small size (average size of 25 nm), high stability, and low clearance rate, making them well-suited for efficient drug delivery in vivo. Importantly, this delivery system showed superior performance compared to traditional carriers like liposomes, offering enhanced tumor suppression with reduced systemic toxicity.

In their study, Lee et al. [117] investigated the efficacy of cancer-specific CPP, specifically BR2, as a delivery vector for vascular endothelial growth factor (VEGF) siRNA in cancer therapy. By forming stable complexes with siRNA, BR2 demonstrated efficient penetration into cancer cells without cytotoxicity to normal cells. The study evaluated the physical parameters of BR2-siRNA complexes, including size, surface charge, and stability, which are crucial for optimal pharmacokinetics and pharmacodynamics. BR2 significantly increased siRNA stability in serum and exhibited higher transfection efficiency and gene silencing in cancer cell lines compared to non-cancer cell lines. The study also compared BR2 to other CPPs, such as R9 and PEI, demonstrating BR2’s superior performance and lower toxicity. Notably, BR2 exhibited a higher transfection efficiency into cancer cells relative to non-cancer cells, affirming its cancer-specificity.

### 4.2. Via Proton Sponge Effect

The proton sponge effect (also known as the pH-buffering effect) is a widely exploited phenomenon in drug delivery and nanoparticle design to facilitate endosomal escape and enhance intracellular delivery of therapeutic agents [118]. It relies on the ability of certain materials to buffer the acidic pH within endosomes, leading to osmotic swelling, rupture, and subsequent release of encapsulated cargo into the cytoplasm [119]. This is primarily attributed to the presence of weakly basic functionalities, such as amines or amidines, within the structure of the delivery vehicle. These functional groups possess pKa values close to physiological pH, enabling them to exist predominantly in a protonated state within the acidic environment of endosomes [120]. As a result, these protonated groups sequester protons from the surrounding medium, leading to an influx of chloride ions to maintain electroneutrality, followed by an influx of water molecules due to osmotic imbalance [121]. The accumulation of chloride ions and water molecules within the endosome results in osmotic swelling and an increase in intraluminal pressure, ultimately leading to the rupture of the endosomal membrane and the release of the encapsulated cargo into the cytoplasm [122]. This process is akin to “sponges” absorbing water and expanding, hence the term “proton sponge.”

In a recent investigation, Kauffman et al. [123] detailed the development of biodegradable poly(amine-co-ester) (PACE) polymers, highlighting their versatile utility in nucleic acid delivery and biomedical contexts. Employing a methodical synthesis approach, the team engineered PACE polymers with customizable attributes, spanning from polyplexes to solid nanoparticles, each imbued with distinct physicochemical properties critical for effective cellular uptake and payload release via the proton sponge effect. The synthesized PACE polyplexes and nanoparticles exhibited advantageous mildly cationic characteristics and nanoscale dimensions, facilitating robust internalization across diverse cell types. Evidenced by heightened fluorescence uptake levels in HEK293, A549, HDF, and NIH 3T3 cells, these delivery vehicles demonstrated pronounced efficacy in traversing cellular membranes. Biocompatibility assessments underscored the excellent safety profile of these delivery vehicles within therapeutic concentration ranges. The PACE polymers were efficient in transporting various nucleic acids—encompassing plasmid DNA, mRNA, and siRNA—into target cells. Remarkably, the transfection efficiencies rivaled or exceeded those of commercial transfection agents (Figure 4A). In nanoparticle form, it facilitated sustained release kinetics, particularly advantageous for larger nucleic acid cargoes such as plasmid DNA and mRNA.

Wang et al. [124] introduced a pH-responsive silica–metal–organic framework hybrid nanoparticle (SMOF NP) engineered to harness the proton sponge effect for efficient delivery of hydrophilic payloads (encompassing small molecule drugs, nucleic acids, and CRISPR-Cas9 genome-editing machinery). Synthesized using a water-in-oil emulsion technique, SMOF NPs exhibited remarkable loading content (>9 wt%) and efficiency (>90%) across various payloads. Through systematic optimization studies, critical parameters influencing payload delivery were identified, including feed ratios of payload to SMOF NP reactants, silica to MOF reactants, and sonication methodologies. Functionalization of SMOF NPs with targeting ligands, such as all-trans retinoic acid, augmented genome editing efficiency within murine retinal pigmented epithelium tissue. In vivo investigations conducted in transgenic mice demonstrated the efficacy of SMOF NPs in facilitating precise genome editing via subretinal injection, thus highlighting their translational significance in advancing precision medicine and gene therapy initiatives.

Yang et al. [125] investigated the use of calcium acetate for remote loading of the weakly acidic drug SN28560 into pH-sensitive liposomes (PSL), offering insights into augmented cytosolic delivery to cancer cells via the proton sponge effect. Their findings revealed a nuanced interplay of factors contributing to enhanced drug loading efficiency, including the establishment of a pH gradient across liposome membranes and the formation of drug-Ca^2+^ complexes. Remarkably, PSL achieved a high drug loading (>30%) and encapsulation efficiency (>95%), outperforming non-pH-sensitive liposomes (NPSL). In vitro studies demonstrated a remarkable decrease in IC_50_ values for PSL-SN25860 compared to NPSL-SN25860 and free drug solutions, underscoring the superior cytotoxicity of PSL formulations. Moreover, tumor accumulation studies revealed substantially higher drug concentrations in PSL-treated groups compared to free drug solutions, highlighting the efficacy of calcium-enabled liposomal delivery. Live-cell imaging provided visual evidence of endosomal disruption induced by calcium-loaded PSL, corroborating the proton sponge effect mechanism (Figure 4B).

**Figure 4 molecules-29-03131-f004:**
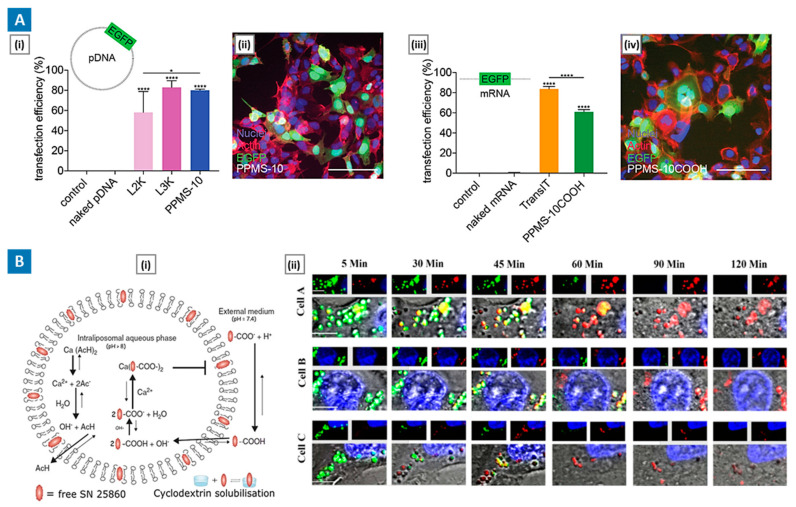
(**A**) Customized nucleic acid delivery using PACE polyplexes that exhibit proton sponge effect. Sub-figure (**i**) shows the efficiency of EGFP plasmid delivery with PACE (PPMS-10) polyplexes compared to Lipofectamine 2000 and 3000 (quantified by flow cytometry). Sub-figure (**ii**) shows a representative fluorescence image of EGFP expression following pDNA delivery by PPMS-10 polyplexes (nuclei are shown in blue, and actin is shown in red). Sub-figure (**iii**) shows the efficiency of EGFP mRNA delivery with different PACE (PPMS-10COOH) polyplexes compared to TransIT (quantified by flow cytometry). Sub-figure (**iv**) shows a representative fluorescence image of EGFP expression following mDNA delivery by PPMS-10COOH polyplexes (nuclei are shown in blue, and actin is shown in red). X. * *p* < 0.05, **** *p* < 0.000. Adapted with permission from [123], Copyright American Chemical Society 2018. (**B**) pH-sensitive liposomes for augmented cytosolic delivery to cancer cells via proton sponge effect. Sub-figure (**i)** shows the mechanism of calcium-enabled remote loading into pH-sensitive liposomes. Sub-figure (**ii**) shows representative images showing PSL containing 500 mM calcium acetate induces rupture of endo/lysosomes from three different representative cells. Here, endo/lysosomes of cells were pre-stained with LysoTracker (green counterstain) and treated with PSL dyed with Rh-PE (red counterstain) and the interaction was monitored by live cell imaging. Pictures of cells at 5 min show that few endo/lysosomes contain liposomes (red signal). Adapted with permission from [125], Copyright Springer Nature 2022.

### 4.3. Via Membrane Fusion

Endosomal escape via membrane fusion represents a pivotal process wherein the endosomal membrane fuses with the surface of nanoparticles, culminating in the liberation of encapsulated cargo into the cytoplasm, thereby enabling access to intracellular targets. This intricate phenomenon is governed by a cascade of events orchestrated by specific membrane-bound proteins or peptides on the nanoparticle surface, which interact with complementary receptors or lipids on the endosomal membrane [24,126]. This interaction initiates a sequence of conformational alterations and structural reconfigurations, ultimately facilitating the integration of the nanoparticle’s constituents with the endosomal membranes. Notably, endosomal escape via fusion is more prominent in membrane/vesicle-like systems [127].

One mechanism by which nanosystems facilitate membrane fusion is through the presentation of fusogenic peptides/proteins on their surface. These fusogenic entities, such as viral fusion peptides or CPPs, possess the capability to engage with lipid components of the endosomal membrane, inducing membrane destabilization or fusion [128]. This phenomenon may be facilitated by variations in pH, alterations in membrane lipid composition, or the presence of specific cellular receptors [129]. Alternatively, nanosystems can harness cellular machinery implicated in physiological membrane fusion processes, such as exocytosis or viral entry pathways [130]. By emulating the mechanisms employed by viruses or intracellular vesicles to fuse with cellular membranes, nanoparticles opportunistically exploit these pathways to effectuate their own endosomal escape and subsequent cytoplasmic delivery [131].

For instance, haemagglutinin is widely employed as a fusogenic agent. It is a peptide found in the influenza virus coat that serves as a fusogenic agent that undergoes a pH-dependent transition from an anionic, hydrophilic coil at pH 7.4 to a hydrophobic helical conformation at the acidic pH within endosomes [132,133]. This structural alteration facilitates the fusion of the viral membrane with the cellular membrane, thereby promoting endosomal escape. Alternatively, the incorporation of fusogenic lipids, such as dioleoylphosphatidylethanolamine (DOPE), is routine in lipid-based nanosystems as “helpers”. It undergoes conformational change upon acidification and promotes a non-lamellar lipid phase change, thereby augmenting the nanosystems’ propensity for membrane fusion-mediated endosomal escape [134].

In an investigation, Pozzi et al. [135] explored the mechanisms underlying the heightened transfection efficiency observed with cholesterol-containing lipoplexes (CCLs). Through meticulous structural characterization utilizing techniques such as dynamic light scattering and synchrotron small angle X-ray scattering, they revealed that the introduction of cholesterol into lipoplexes led to a reduction in the hydration repulsion between lipid membranes, facilitating enhanced membrane fusion with both plasma and endosomal membranes. This fusion mechanism, reminiscent of early stages of viral infection, was found to be directly correlated with the cholesterol content of lipoplexes, with higher cholesterol levels resulting in a more pronounced increase in transfection (Figure 5). Moreover, their study demonstrated that CCLs utilize a cholesterol-dependent macropinocytosis pathway and a temperature-independent mechanism for cellular entry, with the latter showing heightened efficacy with increasing cholesterol content. Additionally, the absence of evidence suggesting preferential metabolic degradation of lipoplexes with high cholesterol content underscores the robustness of CCLs as gene delivery vectors.

A recent study by Gomes et al. [136] represents a significant advancement in cancer therapy, employing long-circulating and fusogenic liposomes loaded with a glucoevatromonoside derivative (SpHL-GEVPG) to induce a potent antitumor response. SpHL-GEVPG’s optimal mean diameter enabled its intravenous administration and tumor accumulation via the enhanced permeability and retention effect. The formulation’s near-neutral zeta potential, facilitated by polyethylene glycol molecules, ensured stability by mitigating vesicle aggregation. High GEVPG entrapment within the lipid bilayer, coupled with a sustained release over 30 days, underscored robust interactions between the compound and liposomal constituents. In vitro cytotoxicity assays against breast and lung cancer cell lines demonstrated substantial reductions in cell viability, with SpHL-GEVPG exhibiting tumor-selective activity. Long-term studies revealed inhibition of surviving tumor cell growth and colony formation, indicative of sustained antitumor efficacy. Importantly, in vivo evaluation in a human lung cancer xenograft model illustrated potent suppression of tumor growth, surpassing the efficacy of paclitaxel treatment.

### 4.4. Via photochemical Disruption

Photochemical disruption exploits principles of photochemistry to induce localized disruption of endosomal membranes, facilitating the release of encapsulated payloads [137]. This process hinges on the utilization of light-responsive materials, including photoactivatable molecules or nanoparticles equipped with photothermal or photodynamic agents. Upon exposure to light of specific wavelengths, these photo responsive components undergo activation, generating reactive species or heat within the endosomal compartment [138,139]. This localized energy release induces structural changes in the endosomal membrane, ultimately destabilizing or rupturing it [140].

Various strategies have been devised to achieve photochemical disruption. One approach involves the use of photoactivatable molecules, such as caged compounds or photocaged peptides, which remain inert until activated by light. Upon irradiation with appropriate wavelength light, these molecules undergo a chemical transformation, releasing bioactive moieties capable of perturbing endosomal membrane integrity [141]. Alternatively, nanocarriers equipped with photothermal or photodynamic agents can be employed. Several photosensitizers, including TPPS_4_-, TPPS_2a_-, AlPcS_2a_-, and dendrimer-based photosensitizers, are primarily localized in the endosomal and lysosomal membranes [142]. Upon light exposure, these photosensitizers induce the formation of reactive singlet oxygen, which has a short lifespan and disrupts the endosomal/lysosomal membrane while leaving organelle contents intact, facilitating delivery to the cytosol [143,144].

The precise spatiotemporal control provided by photochemical disruption offers several advantages for intracellular drug delivery. Selectively irradiating target cells or tissues with light allows researchers to trigger endosomal escape with high spatial resolution, minimizing off-target effects and maximizing therapeutic efficacy [145]. Moreover, the adjustability of light parameters, including intensity, wavelength, and duration, enables precise control over the extent and kinetics of endosomal disruption, further enhancing the adaptability of this approach [146].

Jayakumar et al. [147] introduced a novel approach to augmenting nanoparticle-mediated gene therapy by leveraging a near-infrared (NIR)-light-based nano-platform. Through meticulous synthesis and characterization, core-shell upconversion nanoparticles (UCNs) emitting both UV and visible light upon NIR excitation were developed, enabling simultaneous photocontrolled gene expression and photochemical internalization (Figure 6A). Detailed analysis, including TEM imaging and fluorescence spectroscopy, confirmed efficient loading and sustained release of therapeutic molecules, such as the photosensitizer TPPS2a and photomorpholinos, crucial for gene knockdown. In vitro experiments showcased enhanced endosomal escape and gene knockdown efficacy, while in vivo studies in a murine melanoma model exhibited significant tumor regression with negligible toxicity. Further controls, including treatment with UCNs alone, NIR alone, and UCNs loaded with photomorpholinos but without NIR irradiation, corroborated the specificity and efficacy of the nano-platform. Moreover, tissue analysis revealed lower STAT3 levels in treated mice, affirming the therapeutic potential of the approach.

Nomoto et al. [148] presented a breakthrough in light-responsive gene delivery using a three-layered polyplex micelle (DPc-TPM) nanocarrier platform. By sequentially assembling plasmid DNA (pDNA) and a dendrimeric photosensitizer (DPc) with triblock copolymers, the nanocarrier achieved spatially segregated compartments conducive to efficient gene transduction. Notably, DPc incorporation into the intermediate layer facilitated stable pDNA packaging while preventing oxidative damage during photochemical internalization (PCI). Morphological analyses revealed the structural stability of DPc-TPMs, with TEM confirming rod-shaped pDNA structures within the core. Furthermore, in vitro studies demonstrated multistep DPc and pDNA delivery, with DPc facilitating PCI by translocating to lysosomal membranes in response to acidic conditions (Figure 6B). In vivo experiments validated the efficacy of DPc-TPMs in achieving light-selective gene expression in tumors following systemic administration, showcasing their potential for targeted gene therapy.

In a recent study, Yang et al. [37] showcased a pioneering method to augment siRNA delivery efficacy and its subsequent antitumor effects through photothermal activation-induced endosomal escape. Leveraging melanin as a potent photothermal sensitizer, they engineered melanin-poly-L-lysine (M-PLL) nanoparticles capable of generating localized heat upon exposure to NIR irradiation. These nanoparticles efficiently condensed siRNA via electrostatic interactions, forming stable complexes conducive to intracellular delivery. Following cellular internalization, NIR irradiation triggered the generation of heat, prompting the disruption of endosomal membranes and facilitating siRNA release into the cytoplasm. This photothermally induced endosomal escape significantly enhanced gene silencing efficiency, as evidenced by robust downregulation of target genes both in vitro and in vivo. Notably, the M-PLL/siRNA nanoparticles exhibited exceptional biocompatibility and minimal cytotoxicity, underscoring their potential as safe and effective vehicles for therapeutic siRNA delivery.

### 4.5. Other Endosomal Escape Agents

Scientists have begun to emulate the strategies utilized by viruses and bacteria to facilitate endosomal escape. To date, numerous endosomal escape agents have been isolated or synthesized from various sources. Table 2 delineates several such agents that hold promise for augmenting endosomal escape efficiency.

## 5. Challenges and Future Directions

As the field of nanotechnology-based nanosystems for endosomal escape evolves, several pivotal challenges highlight the complex interplay between innovation and application. Addressing these challenges is crucial for advancing the efficacy and safety of smart nanocarriers in medical applications as well as expanding their usability in clinical settings. One of the primary challenges lies in optimizing the efficiency of endosomal escape across various types of cells and therapeutic agents. Different cell types can exhibit unique endocytic pathways and intracellular processing mechanics, which can significantly affect the efficiency of nanocarriers designed for endosomal escape [169]. Furthermore, therapeutic agents themselves vary in their requirements for release and activity within the cellular environment. To address this, it is essential to develop a deeper understanding of cell-specific endocytic pathways and to tailor nanocarrier designs to accommodate the physicochemical properties of different therapeutic agents. This customization will be pivotal in enhancing the targeted delivery and efficacy of treatments [170].

The scalability of manufacturing smart nanosystems presents another significant hurdle. As these systems transition from laboratory research to industrial-scale production, maintaining consistent quality control, reproducibility, and cost-effectiveness becomes increasingly challenging [171]. The precision required in fabricating nanocarriers often involves complex synthesis processes that are difficult to scale without compromising the functional integrity of the product. Strategies to streamline production, perhaps through automation and process optimization, are critical. These improvements must ensure that large-scale production remains economically viable while adhering to stringent quality standards [172].

Regulatory challenges also play a critical role in the clinical translation of smart nanosystems. The approval process for new nanomedicines involves extensive safety and efficacy testing, which can be both time-consuming and costly [173]. Regulatory bodies often require a comprehensive understanding of a nanocarrier’s behavior in the body, including its biodistribution, biodegradation, and potential off-target effects. Developing frameworks for faster regulatory approvals that do not compromise on safety could accelerate the clinical adoption of innovative nanomedicines [174].

Looking toward future directions, the integration of smart nanocarriers with advanced imaging techniques offers exciting opportunities to enhance real-time monitoring of drug delivery and therapeutic responses. This integration could enable clinicians to track the biodistribution of nanocarriers in the body, assess their therapeutic efficacy, and make real-time adjustments to treatment protocols [175]. Such capabilities would not only improve patient outcomes but also aid in the detailed study of nanocarrier behavior in vivo. The combination of endosomal escape systems with other therapeutic modalities like gene delivery, nanovaccines, and immunotherapy holds substantial promise [176,177,178]. By facilitating efficient intracellular delivery and enhancing cellular uptake, smart nanosystems can significantly amplify the effectiveness of these therapies. This synergistic approach could lead to breakthroughs in treating complex diseases, such as cancer, genetic disorders, and infectious diseases, offering more comprehensive and potent therapeutic solutions [179].

AI and machine learning are poised to transform the design and optimization of nanocarriers. They can automate the optimization of nanocarrier designs, processing large datasets to identify optimal configurations that might not be evident through traditional experimental methods [180]. This includes determining the ideal size, shape, surface chemistry, and functionalization of nanocarriers to maximize their efficacy and minimize toxicity [181,182]. Furthermore, AI models can be trained to predict the behavior of nanocarriers within different biological environments, accounting for the complexities of human biology that are often challenging to replicate in lab settings. This predictive capability is crucial for designing nanocarriers that can effectively navigate the body’s immune responses and biological barriers to deliver drugs precisely where they are needed [183]. Additionally, ML algorithms can be instrumental in the personalization of therapies. By analyzing patient-specific data, such as genetic information, disease progression, and previous treatment responses, AI can help customize nanocarrier systems to individual needs, potentially increasing the success rates of treatments. This personalized approach not only enhances therapeutic outcomes but also reduces the risk of adverse effects, paving the way for more patient-centric therapies [184].

Advancements in targeting capabilities represent a critical area for future research. Developing smart nanosystems capable of dynamically adjusting their targeting in response to changes in the cellular environment would mark a significant advancement. For instance, pH-sensitive nanocarriers can be designed to become more permeable or to release their payload when they enter the more acidic environment of tumor tissues or infected cells [185]. Similarly, temperature-sensitive materials might allow nanocarriers to release their cargo in response to the localized heat of inflamed or tumorous tissues [186]. Another approach could involve the use of molecular recognition elements, such as aptamers or antibodies, that bind selectively to disease-specific markers expressed on the surface of target cells [187]. Additionally, the development of dual or multi-targeted systems could offer even greater precision. These systems would utilize a combination of targeting cues, such as pH sensitivity coupled with a specific receptor-ligand interaction, to ensure delivery only to the intended site. Such adaptive targeting strategies could improve the selectivity and efficacy of nanocarriers, minimizing side effects and enhancing therapeutic outcomes [188].

## 6. Conclusions

This review meticulously examines the evolving landscape of smart nanosystems, highlighting their critical role in enhancing endosomal escape, thereby augmenting the delivery and efficacy of therapeutic agents. Through comprehensive analysis of various mechanisms, such as pore formation, the proton sponge effect, and membrane destabilization, this discussion elucidates how smart nanocarriers can surmount the formidable challenges posed by the endo-lysosomal pathway. The integration of cutting-edge materials and pioneering engineering techniques has culminated in the development of nanocarriers that adeptly navigate cellular environments, consequently boosting therapeutic efficacy and safety. These advancements are not merely incremental; they signify profound strides in our capacity to address diseases at the cellular level. Smart nanosystems have demonstrated their potential across a broad spectrum of applications, from targeted cancer therapies to the delivery of genetic material, showcasing their versatility and revolutionary impact on medical treatments.

Nonetheless, the path toward fully harnessing the potential of smart nanosystems is fraught with challenges. Issues such as scalability of manufacturing, ensuring biocompatibility, and achieving precise targeting continue to present substantial obstacles. Furthermore, the regulatory environment remains complicated, necessitating frameworks that can adapt to the swift progress in nanotechnology while safeguarding patient safety. Future research should concentrate on overcoming these hurdles through interdisciplinary collaboration and innovation. In conclusion, the domain of smart nanosystems for cellular delivery stands at an exhilarating crossroads. The progress observed not only holds promise but also reflects the transformative potential of these technologies. With continued dedication and creativity, smart nanosystems are poised to play an indispensable role in the future of medicine, offering targeted, efficient, and safer therapeutic alternatives to patients globally.

## Figures and Tables

**Figure 1 molecules-29-03131-f001:**
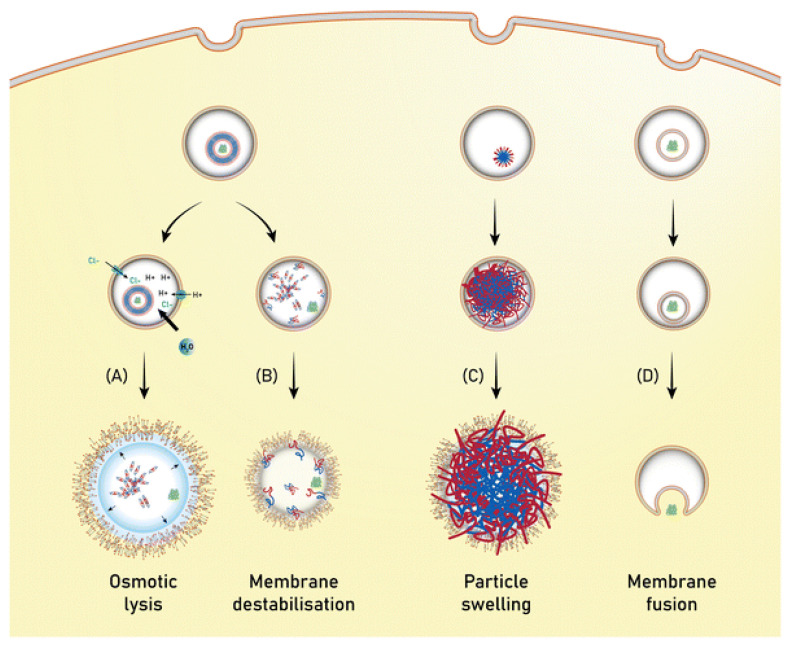
Mechanisms of endosomal escape mediated by the nanovesicular system. (**A**) The “proton sponge” effect occurs when pH-responsive polymersomes (or other polymeric nanocarriers) buffer ions, triggering a rise in osmotic pressure and endosome/lysosome membrane breach. (**B**) Disassembly of pH/reduction-responsive polymersomes into amphiphilic unimers disrupts the endosome/lysosome membrane. (**C**) Swelling of pH-responsive nanocarriers, such as nanoparticles, causes mechanical stress on the membrane. Swelling may also be caused by osmotic rupture or the “proton sponge” effect. (**D**) Fusion of the nanocarrier, usually liposomes, with the endosome/lysosome membrane. (**A**,**B**) are the most often used strategies for polymersome protein delivery. Reproduced with permission from Reference [26].

**Figure 2 molecules-29-03131-f002:**
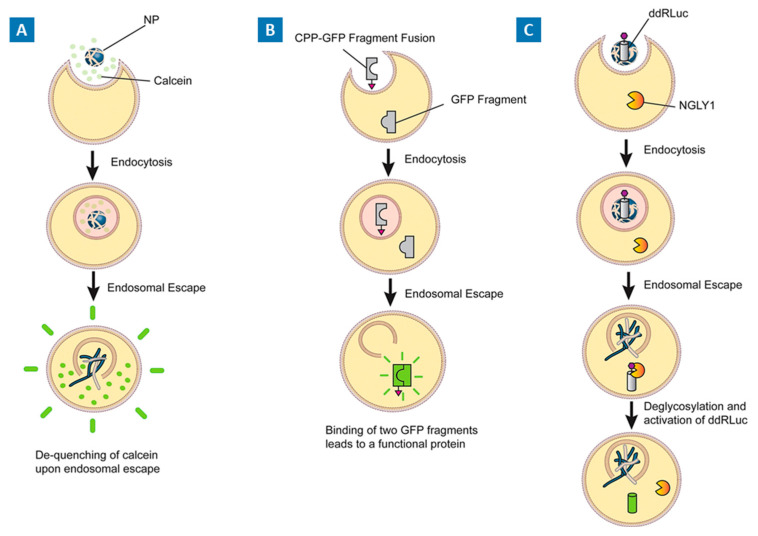
(**A**) Leakage assay. Cells internalize nanoparticles loaded with calcein dye via endocytosis, directing them into endosomal compartments. Calcein, while encapsulated within these endosomes, remains in a quenched state. However, upon nanoparticle-mediated endosomal escape and subsequent release into the cytosol, calcein becomes unquenched, thus becoming detectable. (**B**) Complementation assays. A split GFP complementation assay presents an adaptable method for quantifying the endosomal escape of CPPs. This technique involves the fusion of the CPP with a GFP fragment. Successful endosomal escape leads to the reconstitution of a functional GFP reporter. (**C**) Cytosolic activation assays. Upon intracellular delivery, a deactivated ddRLuc probe is introduced into the cellular milieu. Subsequent release into the cytosol prompts the NGLY1 enzyme to catalyze the reconstitution of the functional luciferase, culminating in the manifestation of a luminescent signal. Adapted with permission from [28], copyright Elsevier 2021.

**Figure 3 molecules-29-03131-f003:**
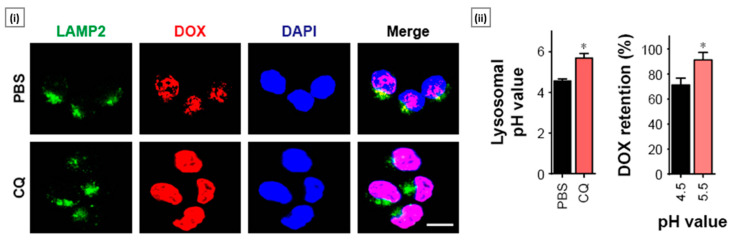
CQ exhibited sensitizing effects on ovarian cancer cells by reversing the sequestration of chemotherapeutic agents (DOX) within lysosomes. Sub-figure (**i**) shows two photon confocal microscope images of A2780 cells that were treated with DOX (4 μM, 4 h) pretreated with or without CQ (10 μM, 2 h), followed by LAMP2 staining (scale bar: 10 μm). Sub-figure (**ii**) shows the values of lysosomal pH values of A2780 cells treated with or without CQ (10 μM) for 2 h along with metabolisms of DOX under different pH conditions (examined by HPLC). * *p* < 0.05. Adapted with permission from [116], copyright Dove Press Ltd. 2018.

**Figure 5 molecules-29-03131-f005:**
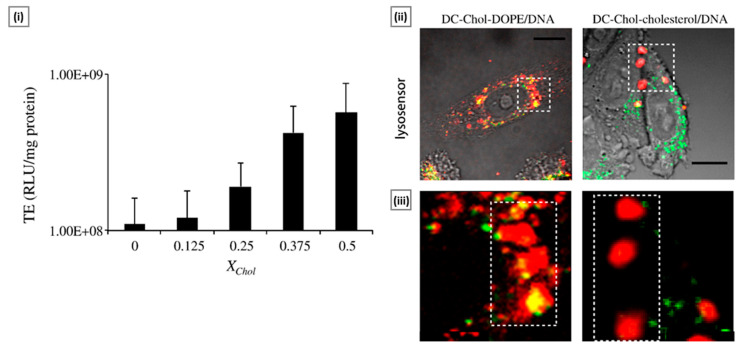
Transfection efficiency boost of cholesterol-containing lipoplexes. Sub-figure (**i**) shows the transfection efficiency (TE) in RLU per mg of cellular proteins of DC-Chol–DOPE–cholesterol/DNA lipoplexes as a function of increasing molar fractions of cholesterol (XChol = 0, 0.125, 0.25, 0.375, 0.5). TE increases over about one magnitude with an increasing molar fraction of cholesterol. Sub-figure (**ii**) shows the colocalization of DC-Chol–DOPE/DNA and DC-Chol–cholesterol/DNA signals (red) with Lysosensor (lysosome marker, green), after 3 h of lipoplex treatment. Both lipoplex formulations are able to avoid lysosomal entrapment. Sub-figure (**iii**) shows DC-Chol–cholesterol/DNA-lipoplex-induced structural features that arise from extensive membrane reorganization. Adapted with permission from [135], copyright Elsevier 2012.

**Figure 6 molecules-29-03131-f006:**
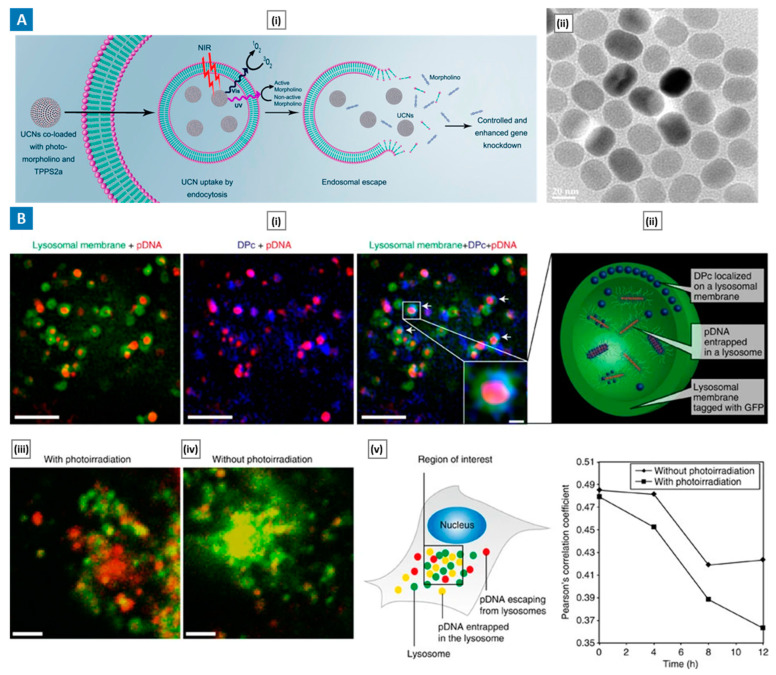
(**A**) NIR-based nano-platform to boost endosomal escape. Sub-figure (**i**) shows a schematic depiction of NIR-mediated photochemical disruption to facilitate endosomal escape. Sub-figure (**ii**) shows a TEM image of the NIR-to-UV/Vis core-shell UCNs. Adapted with permission from [147], Copyright American Chemical Society 2018. (**B**) Three-layered polyplex micelle for light-induced systemic gene transfer. Sub-figure (**i**) shows super-resolution microscopic images of HeLa cells incubated with DPc-TPMs for 6 h. White arrows indicate the colocalization of lysosomal membranes and DPc (scale bar: 2 μm; scale bar for magnified image: 200 nm). Sub-figure (**ii**) shows a schematic view of the assumed localization of DPc and pDNA in the lysosomal compartment. Sub-figure (**iii**) shows confocal imaging of the subcellular distribution of the micelles 15.5 h after photoirradiation. Lysosomal membranes were tagged with GFP (green). Cy3-labeled pDNA is shown in red (scale bar: 2 μm). Sub-figure (**iv**) shows confocal imaging of the subcellular distribution of DPc-TPMs without photoirradiation (scale bar: 2 μm). Sub-figure (**v**) shows a schematic diagram and corresponding quantification of Pearson’s correlation coefficient between lysosomes–GFP and Cy3-labelled pDNA. Adapted with permission from [148], copyright Springer Nature 2014.

**Table 1 molecules-29-03131-t001:** Pharmacological inhibitors available for endosomal escape of nanosystems.

Inhibitor	Mechanism/Description	Limitations	Experimental Observations	Ref.
Chlorpromazine	• CME inhibition. • Sequesters clathrin and AP2 complexes away from the cell membrane and directs them towards endosomal compartments.	It can impact clathrin-independent cellular pathways and reduce cellular viability.	The inclusion of chloroquine resulted in an enhancement of transfection efficiency associated with the PEI-based polyplexes. A notable proportion of the PEI polyplexes underwent trafficking via acidifying endosomes to lysosomes. The buffering effect mediated by chloroquine synergized vesicular escape and subsequent transfection mediated by these polyplexes.	[76]
Methyl-β-cyclodextrin (MβCD)	• Caveolae-mediated inhibition. • Forms complexes with cholesterol within the cell membrane, leading to its depletion.	Influences/alters CME and various other endocytic pathways.	The internalization pathway of CPP-functionalized iron oxide nanoparticles was probed using sodium azide/2-deoxy-D-glucose, known as energy inhibitors, dynasore (a dynamin inhibitor), and MβCD. Across all treatment groups, a notable decrease in nanoparticle uptake was observed, implicating the involvement of clathrin- and caveolae-mediated endocytosis pathways. However, treatment with Pitstop 2, a clathrin inhibitor, did not yield a significant impact on nanoparticle uptake. This discrepancy suggests potential non-specific effects of chemical inhibitors	[77]
Bafilomycin A1	• Endosome maturation inhibition. • Blocking of vacuolar proton ATPases.	Can induce cytoplasmic acidification through proton accumulation.	The dynamics and mechanisms of PEI-mediated plasmid delivery were evaluated using Bafilomycin A1. Pre-treatment decreased transfection 30-fold, whereas the addition of the Bafilomycin A1 4 h after PEI treatment only decreased transfection efficiency by 33%, suggesting that the majority of endosomal escape occurs before 4 h.	[72]

**Table 2 molecules-29-03131-t002:** Endosomal escape agents.

Type	Agent	Mechanism	Ref.
Virus-derived proteins/peptides	Poly(L-lysine)	Membrane fusion	[149]
	diINF-7	Membrane fusion	[150]
	Penton base	Pore formation	[151]
	gp41	Unclear	[152]
	L2 from Papillomavirus	Membrane fusion	[153]
Bacteria-derived proteins/peptides	Listeriolysin O toxin	Pore formation	[154]
	Pneumococcal pneumolysin	Pore formation	[155]
	Diphtheria toxin	Membrane fusion	[156]
	*Pseudomonas aeruginosa* Exotoxin A	Pore formation	[157]
Mammalian proteins/peptides	Melittin	Pore formation	[158]
	Human calcitonin-derived peptide	Unclear	[159]
Synthetic peptides	Glycoprotein H from herpes simplex	Membrane fusion	[160]
	KALA	Membrane fusion	[161]
	GALA	Membrane fusion	[162]
	Bovine prion protein	Pore formation	[163]
	Poly(L-histidine)	Proton sponge effect	[164]
	Proline-rich peptide	Membrane fusion	[165]
Chemicals	Ammonium chloride	Proton sponge effect	[166]
	Poly(propylacrylic acid)	Proton sponge effect	[167]
	Poly(amidoamine)	Proton sponge effect	[168]

## Data Availability

Not applicable.

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
