# Peer review of "Achieving Endo/Lysosomal Escape Using Smart Nanosystems for Efficient Cellular Delivery"

_molecules, 2024, doi:10.3390/molecules29133131_

Round 1

Reviewer 1 Report

Comments and Suggestions for Authors

There are a variety of parameters affecting the endo/lysosomal escape efficiency of nanomaterials that are internalized by target cells. This review highlights current strategies for breaking endo/lysosomal barriers and discusses challenges of nanomedicines. The authors also point out the most recent advancements by AI and machine learning in customizing nanocarrier systems. Overall, the review is concise and well organized. The following questions should be addressed before acceptance by Molecules:

1.     In the title, “endosomal escape” might be replaced by “endo/lysosomal escape”.

2.     How about endo/lysosomal redox environment and should the redox environment be considered for endo/lysosomal escape?

3.     May the authors discuss more about strategies of endo/lysosomal escape in protein delivery?

4.     Is there any potential endo/lysosomal biomarker that is utilized to discriminate cancer cells from healthy cells?

5.     Are in vitro endo/lysosomal escape results usually correlated with in vivo performance? If not, why?

Author Response

  1. In the title, “endosomal escape” might be replaced by “endo/lysosomal escape”.

Ans: we appreciate the reviewer’s opinion. As suggested, the required changes have been made in the revised manuscript

  1. How about the endo/lysosomal redox environment and should the redox environment be considered for endo/lysosomal escape?

Ans: Endo/ lysosomal redox environment may affect the fate of the nano delivery system and encapsulated therapeutic agents. This redox environment may change the efficacy of the drug molecules. Further, drug release mechanisms can be altered due to this environment. Moreover, the toxicity and distribution of drugs or vehicles can be changed. Therefore, the endo/lysosomal redox environment should be considered during the designing of the drug delivery system.

  1. 3. May the authors discuss more about endo/lysosomal escape strategies in protein delivery?

Ans: We appreciate the reviewer's suggestion. However, scientists have been utilizing the same strategies of small molecules for protein and peptide-based drugs.

  1. Is there any potential endo/lysosomal biomarker that is utilized to discriminate cancer cells from healthy cells?

Ans: There are several biomarkers have been studied such as Lysosome-associated membrane proteins, lysosomal proteins, glycans and glycosylation patterns, endosomal markers such as Rab proteins and lysosomal enzyme (cathepsin) to discriminate cancer cells from healthy cells

  1. Are in vitro endo/lysosomal escape results usually correlated with in vivo performance? If not, why?

Ans: It does not correlate with in vivo performance. There are several reasons:

  1. We can't mimic the exact in vivo environment with in vivo study model such as biochemical reactions in the body
  2. Pharmacokinetics factors such as distribution of nano vehicles in circulation or organ, metabolic activities, and clearance of these nano vehicles can't be controlled in an in vitro study model.

Reviewer 2 Report

Comments and Suggestions for Authors

Desai et al. have given brief overview of significance of endosomal entrapment in cellular delivery. Further, authors have represented several assays to study the endosomal escape and uptake effectively. Therefore, present work would be beneficial to nano delivery scientist. Hence, reviewer would like to recommend this article in molecule journal in the present form. However, few improvements need to be addressed before accepting manuscript for publication.

1. Include diagrammatical representation for endosomal escape in the introduction

2.Verify whole manuscript for consistent abbreviations

3.Include few current references in the revised manuscript

Author Response

Desai et al. have given brief overview of significance of endosomal entrapment in cellular delivery. Further, authors have represented several assays to study the endosomal escape and uptake effectively. Therefore, present work would be beneficial to nano delivery scientist. Hence, reviewer would like to recommend this article in molecule journal in the present form. However, few improvements need to be addressed before accepting manuscript for publication.

Ans: We appreciate the reviewer's positive comments toward our work.

  1. Include diagrammatical representation for endosomal escape in the introduction

Ans: As per the reviewer’s suggestion, required diagram has been included in the revised manuscript and highlighted in yellow color.

2. Verify whole manuscript for consistent abbreviations

Ans: As suggested, the whole manuscript has been verified for abbreviations consistency.

3.Include few current references in the revised manuscript

Ans: As suggested. A few more references have been included in the revised manuscript.